# Prevalence of Bruxism Among Young Adult Energy Drink Consumers in Jordan

**DOI:** 10.3390/jcm15010203

**Published:** 2025-12-26

**Authors:** Aseel M. Sharaireh, Musab M. Alkaabneh, Hamzeh E. Alsaket, Hamza I. Abdelhaleem, Amr I. Hammad, Noor H. Ismail, Islam Abd Alraheam, Sanaa Aljamani, Leena Smadi, Yazan Hassoneh, Mohammad A. AL-Rabab’ah

**Affiliations:** 1Restorative Dentistry Department, School of Dentistry, University of Jordan, Amman 11942, Jordan; musabkaabneh@gmail.com (M.M.A.); hamzehalsaket21@gmail.com (H.E.A.); hamzaimadsalah@gmail.com (H.I.A.); amrimadhammad@gmail.com (A.I.H.); noor.ismail@ju.edu.jo (N.H.I.); i.abdalraheam@ju.edu.jo (I.A.A.); s.aljamani@fisjo.onmicrosoft.com (S.A.); l.smadi@ju.edu.jo (L.S.); 2Department of Oral and Maxillofacial Surgery, Oral Medicine and Periodontology, School of Dentistry, University of Jordan, Amman 11942, Jordan; yazan@ju.edu.jo

**Keywords:** bruxism, energy drinks, caffeine, stress, young adults, TMJ

## Abstract

**Background/Objectives:** This study examined the association between energy drink consumption and self-reported bruxism behaviors and temporomandibular joint (TMJ) pain among young adults in Jordan. **Methods**: A cross-sectional, self-administered survey was distributed to young adults aged 18–30. After applying predefined exclusion criteria, the analytic sample for the TMD-related analyses comprised *n* = 1373 participants. The questionnaire captured demographics, frequency and duration of energy drink consumption, self-reported bruxism behaviours (clenching, grinding, bracing, thrusting), TMJ pain symptoms adapted from DC/TMD screening items, and psychological stress measured by the validated Perceived Stress Scale (PSS-10). Questionnaire reliability was assessed in a pilot (*n* = 20) using Cohen’s Kappa. Descriptive statistics, chi-square tests and multivariable logistic regression (adjusting for age, sex, smoking and stress) were used to evaluate associations. **Results**: Among participants, 309 (22.5%) reported daily energy drink use. Self-reported bruxism behaviors were present in 19.4% (*n* = 60) of consumers, with an additional 26.9% suspecting these behaviors. TMJ pain on awakening was reported by 41.1% (*n* = 127) of consumers. Energy drink consumption was significantly associated with higher odds of self-reported bruxism behaviors (χ^2^ = 115.6, *p* < 0.001). In multivariable analyses, daily consumption remained independently associated with bruxism (adjusted OR 1.9; 95% CI 1.3–2.7; *p* = 0.001). Higher consumption frequency was also linked to an increased number of oral health symptoms. **Conclusions**: after adjusting for key confounders, energy drink consumption was associated with greater prevalence of self-reported bruxism behaviors and TMJ pain among young Jordanian adults. These findings emphasize the potential oral health risks of habitual energy drink use and underscore the need for targeted education and preventive strategies in this population.

## 1. Introduction

Bruxism is currently defined as a repetitive jaw-muscle activity characterized by clenching, grinding, bracing or mandibular thrusting and may occur during wakefulness (awake bruxism, AB) or during sleep (sleep bruxism, SB) [1,2,3]. This behavioral perspective emphasizes bruxism as a motor activity with possible clinical consequences rather than a disease per se. Objective assessment may require electromyography (EMG) or polysomnography (PSG), while standardized subjective tools such as the Oral Behavior Checklist (OBC) and the Standardized Tool for the Assessment of Bruxism (STAB) enable structured self-report assessment [4].

Bruxism is multifactorial: psychological factors (stress, anxiety), sleep disturbances, and stimulant exposure are frequently implicated [5,6]. Awake behaviors (clenching/bracing) often relate to daytime stress and coping styles, whereas sleep bruxism has stronger links to sleep architecture and arousal mechanisms [7]. Recent studies describe gradient relationships between psychological scores and awake bruxism frequency [6,8].

Energy drinks contain biologically active constituents such as caffeine and taurine and high sugar content; these components influence arousal, autonomic tone and sleep quality [9,10]. Caffeine increases central nervous system activity through antagonism of adenosine receptors and may affect motor excitability [9,11]. Taurine can modulate calcium handling in muscle, and sugar additives may contribute to systemic inflammatory responses [12,13]. These physiological effects provide a plausible pathway by which frequent energy drink intake could modulate bruxism behaviours via hyperarousal, sleep fragmentation and altered neuromuscular excitability.

The condition has a complex and not yet fully understood etiology [14]. Several risk factors have been linked to bruxism, including stress, certain oral habits, and the consumption of beverages containing ingredients that stimulate the nervous system, possibly through disturbances in the neurotransmitter gamma-aminobutyric acid (GABAergic) and glutamatergic systems of the brain [15].

Bruxism prevalence varies across different populations. Among children, sleep bruxism (SB) is more commonly reported than in adults [16], with prevalence estimates ranging from 13% to 49% [17]. A large-scale epidemiological study on oral health in the general Dutch adult population, which surveyed 1209 individuals, found that AB was present in 5.0% of participants, while SB was observed in 16.5% [18]. The prevalence of AB was recorded at 6.5%, 7.8%, 4.0%, 3.2%, and 3.0% across five age groups (25–34, 35–44, 45–54, 55–64, and 65–74 years, respectively). Meanwhile, SB was reported at rates of 20.0%, 21.0%, 16.5%, 14.5%, and 8.3% in the same groups. Women reported both AB and SB more frequently than men, with these differences being statistically significant [18].

There has been an observed increase in coffee and energy drink consumption, which has been linked to higher stress levels. Reports indicate that the number of energy drink consumers is rising significantly. For example, in the United States, consumption increased from 0.5% to 5.5% in less than 15 years [19]. Notably, although stress may contribute to the initial rise in energy drink consumption, many individuals continue using these beverages even after stress levels subside.

As energy drink consumption continues to rise, it is essential to examine its key components, particularly caffeine, which plays a significant role in stimulating the nervous system and influencing various physiological processes. Caffeine is a central nervous system stimulant that enhances alertness by blocking adenosine receptors and affecting calcium levels in muscle fibers [9]. It has a half-life of 4–6 h and is commonly found in coffee, tea, chocolate, and energy drinks, with a safe daily intake of around 400 mg [20,21,22]. It is well known that high doses may lead to side effects such as insomnia and muscle spasms [23].

Sugars, particularly high fructose corn syrup in energy drinks, can contribute to health issues like atherosclerosis and chronic inflammation [12,24]. Taurine, an amino acid found in meats and seafood, supports calcium-dependent muscle contraction and has neuroprotective effects [13,25,26,27]. Each energy drink typically contains 600–1000 mg of taurine [28]. Caffeine metabolism involves its breakdown into paraxanthine, theobromine, and theophylline, with only a small percentage excreted in urine [29,30,31]. Studies suggest caffeine may inhibit glucose transporter 1 (GLUT1), leading to increased blood glucose and lipid metabolism [32,33].

The interaction between caffeine and taurine affects cardiovascular function, diuresis, and muscle contraction [34]. While caffeine raises blood pressure, taurine may help lower it [10,35,36]. Research also suggests taurine enhances caffeine-induced muscle contraction by promoting calcium release [37,38].

Despite widespread consumption of these beverages among young adults, direct empirical evidence linking energy drink use to bruxism is limited. This study therefore aimed to examine associations between energy drink consumption, self-reported bruxism behaviors, and TMJ pain in a large sample of Jordanian young adults. We hypothesized that higher frequency of energy drink consumption would be associated with increased prevalence of self-reported bruxism behaviors and TMJ symptoms, independent of stress level and other confounders.

## 2. Materials and Methods

### 2.1. Study Design and Participants

A cross-sectional survey was conducted among Jordanian residents aged 18–30. The questionnaire was distributed electronically between April 2024 and October 2024. The sample size was calculated based on the estimated population of the 18–30 age group in Jordan, approximated at 1.5 million individuals (Population estimates were obtained from the United Nations World Population Prospects https://population.un.org/wpp/) (Accessed on 4 June 2023). Using a 95% confidence level, a margin of error of ±2.5%, and assuming maximum variability (*p* = 0.5), the required sample size was determined to be approximately 1600. However, the study employed a sample size of 1500 participants, which is sufficient to provide reliable and representative results for this population segment. The survey was distributed to 2000 participants, and 1597 completed responses were received, resulting in a response rate of approximately 79.85%. Pre-specified exclusion criteria (age > 30, consumption of >2 cups of coffee/day, diagnosed obstructive sleep apnea, ADHD, GERD, epilepsy, self-reported substance/alcohol use, and very high stress as measured by PSS) were applied to produce the analytic sample for TMD-related analyses (*n* = 1373).

Ethical approval was obtained from the Institutional Review Board at the University of Jordan (Decision No. 46-2023); informed consent was obtained from all participants.

### 2.2. Questionnaire and Measures

The bespoke questionnaire included four sections: demographics; behavioral factors (energy drink and coffee consumption, smoking, alcohol); self-reported bruxism behaviors and TMJ symptoms; and the validated PSS-10 for perceived stress [39]. Self-reported bruxism items covered clenching, grinding, bracing and thrusting and included frequency anchors (never, sometimes, often, always). TMJ pain items were adapted from the DC/TMD screening questions to capture pain presence and timing (e.g., pain upon waking).

Participants reported coffee consumption in terms of the number of cups per day. Although cup volume and caffeine content were not directly measured, previous research in Jordan has shown that a typical cup of brewed coffee contains approximately 113–247 mg of caffeine depending on type and preparation method [40]. This range provides context for interpreting the cutoff of >2 cups per day used to control for additional caffeine intake.

Participants consuming up to two cups of coffee per day were included in the analyses; therefore, some individuals may have consumed both coffee and energy drinks, but overlapping caffeine intake from these sources was not analyzed separately.

### 2.3. Pilot Testing and Reliability

A pilot test (*n* = 20) assessed test–retest reliability (two administrations separated by one week) using Cohen’s Kappa. Kappa values for key items were daily coffee consumption (κ = 0.77), energy drink cans/day (κ = 0.73), bruxism item (κ = 0.91) and pain upon waking (κ = 0.51), supporting acceptable reliability for the instrument.

### 2.4. Statistical Analysis

Data were analyzed using IBM SPSS Statistics v.26. Descriptive statistics summarized sample characteristics. Bivariate associations were examined with chi-square tests for categorical variables and t-tests for continuous measures where appropriate. Multivariable logistic regression models were constructed with self-reported bruxism behaviors (yes/no) and TMJ pain (yes/no) as dependent variables. Key covariates included age, sex, smoking status, and PSS score. Significance was set at *p* < 0.05.

## 3. Results

### 3.1. Sample Characteristics

The demographic characteristics of all 1597 participants in the study are summarized in Table 1, including participants’ occupation, age, and gender distribution. This overview provides context for the population included in the survey before applying the pre-specified exclusion criteria for stress, age, caffeine intake, medical conditions, and substance use. Most participants were aged 18–22 (88%) and were university students (92.3%). Specifically, 45.2% were aged 18–20 years, and 42.8% were aged 20–22 years. The sample included 57.5% female and 42.5% males, reflecting a slightly higher representation of women in the study population. Although the overall population in Jordan shows a slight male predominance, with approximately 106 males per 100 females (populationpyramids.org (https://www.populationpyramids.org/jordan) (Accessed on 10 November 2025)), female students tend to represent a majority in higher education, which is consistent with the gender distribution observed in this sample.

Of the 1597 participants included in the survey, stress levels assessed using the PSS were categorized as follows: low (*n* = 194; 12.1%), moderate (*n* = 1239; 77.6%), and high (*n* = 164; 10.3%) (Table 2). In accordance with the pre-specified protocol, participants classified as high stress (Score = 2) were excluded from the TMD-specific analyses to minimize confounding related to stress-induced pain perception and to better isolate the potential independent association with energy drink consumption. Thus, 164 high-stress individuals were removed prior to further analysis.

In addition to stress-related exclusions, the following groups were also excluded: 40 individuals older than 30 years of age; 204 participants who reported consuming more than two cups of coffee per day, to reduce confounding from high baseline caffeine intake from other dietary sources; and 224 participants with relevant medical conditions, including obstructive sleep apnea (77) and ADHD (152), GERD (71), and epilepsy (17). Furthermore, participants who reported substance use were excluded, including 21 reporting drug use and 53 reporting alcohol consumption.

It is important to note that some individuals meet more than one exclusion criterion. After applying all exclusions, a refined subset of 1373 participants were used for the TMD-specific and energy drink-related analyses to ensure greater validity of the findings.

### 3.2. Energy Drink Consumption

After applying the exclusion described, 309 (22.5%) reported daily energy drink consumption (Table 3). The duration of consumption varied: 28.3% had consumed them for 1–2 years, 19.7% for six months, and a smaller proportion for more than four years (Table 2).

### 3.3. Bruxism Behaviours and TMJ Symptoms

Self-reported bruxism behaviours were reported by 19.4% (*n* = 60) of energy drink consumers; an additional 26.9% suspected they had bruxism. Among consumers, 41.1% (*n* = 127) reported TMJ pain upon awakening. Chi-square tests indicated significant associations between daily energy drink consumption and self-reported bruxism behaviours (χ^2^ = 115.6, *p* < 0.001) and TMJ pain (χ^2^ = 9.79, *p* = 0.0017).

### 3.4. Multivariable Models

In logistic regression adjusting for age, sex, smoking and PSS score, daily energy drink consumption was independently associated with self-reported bruxism behaviors (adjusted OR 1.9; 95% CI 1.3–2.7; *p* = 0.001). Similar models for TMJ pain showed attenuated but directionally consistent associations (adjusted OR 1.4; 95% CI 0.98–2.1; *p* = 0.06). Higher consumption frequency was associated with a greater number of reported oral health symptoms (χ^2^ = 11.74, *p* = 0.0083) (Figure 1), while duration of consumption did not reach statistical significance in relation to symptom complexity (χ^2^ = 29.13, *p* = 0.085) (Figure 2).

## 4. Discussion

This study identified a robust association between frequent energy drink consumption and higher prevalence of self-reported bruxism behaviors and TMJ pain in a large sample of Jordanian young adults. These findings align with the hypothesized pathway whereby stimulant intake (caffeine, taurine, sugar) increases arousal and motor excitability, potentially facilitating daytime bracing/clenching and nocturnal grinding behaviors [9,10,12]. Interpretations in this discussion are based on the study findings and previously published evidence; personal clinical experience has not been included.

Importantly, we adopted contemporary bruxism terminology and emphasize that our outcome measures reflect self-reported behaviors rather than instrumentally confirmed diagnoses [1,3]. Self-report has recognized limitations: individuals may under- or over-report behaviors, and the absence of EMG/PSG prevents classification as definite bruxism. Nevertheless, structured self-report instruments remain useful for large epidemiological studies and for hypothesis generation.

In the present study, bruxism was examined as a self-reported behavioural construct rather than a clinically confirmed diagnosis. Individuals reporting “suspected” bruxism were considered alongside those reporting bruxism behaviours, as both reflect perceived parafunctional activity commonly captured in population-based research. Accordingly, the observed associations reflect patterns of self-reported bruxism behaviours and are not dependent on diagnostic certainty.

Exclusion of individuals with high PSS scores reduced confounding by severe stress, but it also restricts generalizability to lower/moderate stress populations. In everyday settings, energy drink consumption frequently co-occurs with elevated stress levels, particularly among young adults facing academic or occupational demands [41]. As a result, the observed associations may represent conservative estimates of the true relationship. Stress may function not only as a confounder but also as a potential mediator or moderator [42], whereby stimulant intake could exacerbate stress-related neuromuscular activity and parafunctional behaviors such as bruxism. The cross-sectional design precludes causal inference: energy drink use may be a cause, consequence or correlate of bruxism behaviors (for example, as a coping mechanism for stress).

Nonetheless, the persistence of statistically significant associations despite this conservative approach suggests that energy drink consumption may operate as an independent correlate of bruxism behaviours. This adds to a growing body of evidence showing that behaviours involving stimulants (such as caffeine) may interact with neurochemical pathways implicated in bruxism, including dopaminergic, glutamatergic and GABAergic circuits [15]. Experimental studies have demonstrated that caffeine influences central dopaminergic transmission and increases neuromuscular excitability, providing a physiological rationale for the increased jaw-muscle activity observed in individuals consuming stimulants [11,20,23].

In addition to caffeine’s effects, taurine, another prominent constituent of energy drinks, may also influence neurochemical and neuromuscular activity relevant to bruxism. Taurine plays a role in modulating intracellular calcium handling and can influence excitation-contraction processes in skeletal muscle, including jaw musculature [27]. It also interacts with inhibitory neurotransmitter systems such as GABAergic and glycinergic pathways, which may affect motor control and muscle tone [43]. Although taurine alone may not directly increase motor excitability, when combined with caffeine’s stimulatory effects on the central nervous system, they may exert additive or synergistic influences on neuromuscular pathways [44]. Such interactions could contribute to enhanced jaw-muscle activity and parafunctional behaviors observed among frequent energy drink consumers [45].

Strengths of this study include a large sample, use of a validated stress instrument, pilot reliability testing, and pre-specified exclusion criteria. Key limitations are reliance on self-report for bruxism and TMJ symptoms, potential sampling bias inherent to electronic surveys, and lack of objective sleep measures. Despite these limitations, the observed association merits further investigation given the public health relevance of energy drink consumption in young populations.

The cross-sectional design limits causal inference. Energy drink consumption may be a cause, consequence or behavioural correlation of bruxism. For example, individuals with poor sleep due to bruxism-related micro-arousals may increase energy drink consumption as a compensatory strategy to counter daytime fatigue, creating a feedback loop.

Additionally, the study sample consisted predominantly of young adults aged 18–22 years and excluded individuals with several medical issues, which affects the generalizability of the findings. The results may therefore not be directly applicable to older populations, individuals with chronic health conditions, or populations with different lifestyles or cultural characteristics related to caffeine consumption and stress exposure.

Bruxism behaviors were assessed using self-reported measures, which may be subject to recall or reporting bias; however, self-report questionnaires are widely used and considered appropriate in large-scale epidemiological studies where objective monitoring is not feasible.

Future research would benefit from longitudinal methods, ecological momentary assessment (EMA), and ambulatory EMG or smartphone-based bruxism monitoring devices, which have shown reliability in capturing real-time awake bruxism behaviours [5,6,8]. These approaches can clarify temporal ordering and identify whether energy drink intake predicts later changes in bruxism frequency or vice versa. Future longitudinal research using ecological momentary assessment (EMA) and ambulatory EMG would clarify temporal relationships [5,8].

Clinicians assessing patients with bruxism or TMJ pain should routinely enquire about stimulant intake, particularly energy drinks, as part of their behavioural and lifestyle assessment. Counselling on reducing excessive consumption may assist in managing symptoms for susceptible individuals. At the public health level, awareness campaigns could incorporate potential neuromuscular and oral risks associated with high-frequency energy drink intake, complementing existing messaging on cardiovascular, metabolic and sleep-related harms.

## 5. Conclusions

In conclusion, after adjusting for relevant confounders, frequent energy drink consumption was significantly associated with higher odds of self-reported bruxism behaviours and TMJ pain among young adults in Jordan. These findings suggest that energy drink use may independently contribute to neuromuscular hyperactivity in the masticatory system. Future longitudinal studies incorporating objective measures such as EMG or polysomnography are needed to clarify causal relationships and elucidate the underlying physiological mechanisms. Clinicians and public health practitioners should consider addressing high-frequency energy drink consumption as a modifiable risk factor for bruxism and TMJ-related symptoms.

## Figures and Tables

**Figure 1 jcm-15-00203-f001:**
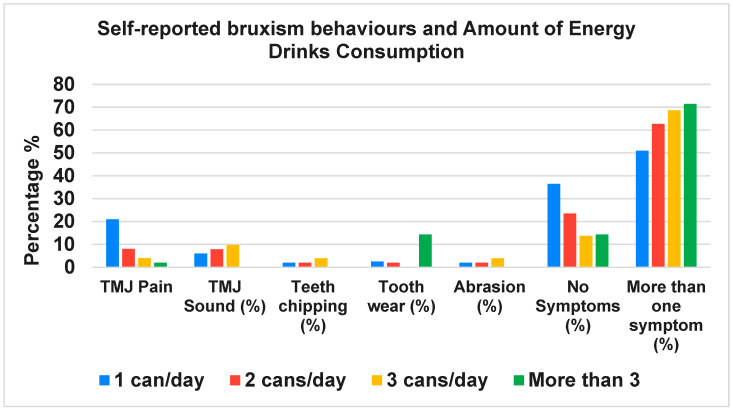
Self-reported bruxism behaviours by amount of energy drink consumed.

**Figure 2 jcm-15-00203-f002:**
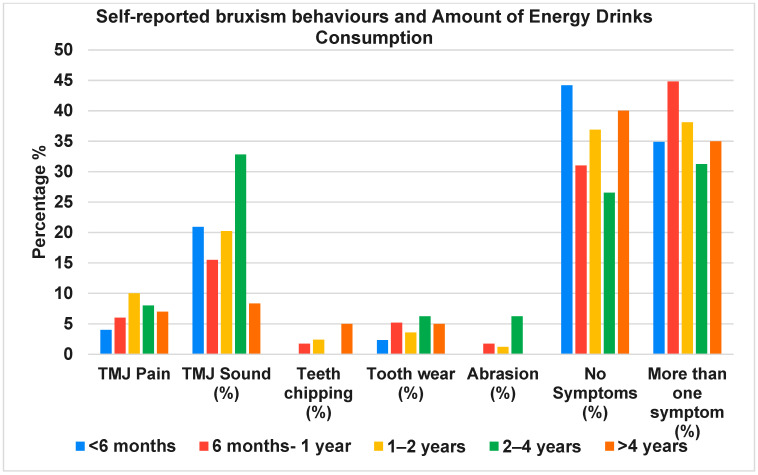
Self-reported bruxism behaviours by duration of energy drink use.

**Table 1 jcm-15-00203-t001:** Demographics of study sample.

Characteristic	Category	Frequency (*n*)	Percent (%)
Occupation	University Student	1474	92.3
	Employee in Private Sector	36	2.3
	Employee in Public Sector	28	1.8
	Secondary School Student	28	1.8
	Others	31	1.9
Age (years)	18–20	722	45.2
	20–22	677	42.4
	22–24	117	7.3
	24–26	23	1.4
	26–28	11	0.7
	28–30	7	0.4
	>30	40	2.5
Gender	Female	921	57.7
	Male	676	42.3
Total	–	1597	100.0

**Table 2 jcm-15-00203-t002:** Self-reported Stress Levels (Validated Scale).

Stress Score	Description	Frequency	Percentage
0	Low Stress	194	12.1%
1	Moderate Stress	1239	77.6%
2	High/Severe Stress	164	10.3%
Total		1597	100.0%

**Table 3 jcm-15-00203-t003:** Distribution of Energy Drink Consumption and Duration.

Measure	Category	*n*	%
Consumption Status (n = 1373)	No	1064	77.5
	Yes	309	22.5
Duration of Consumption (Yes respondents only, n = 309)	< 6 months	43	13.9
	6 months–1 year	58	18.8
	1–2 years	84	27.2
	2–4 years	64	20.7
	> 4 years	60	19.4

## Data Availability

The datasets generated by the survey research during the current study are available in the Dataverse repository, https://doi.org/10.7910/DVN/ZE290H (created 1 August 2025).

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
