# Peer review of "Prevalence of Bruxism Among Young Adult Energy Drink Consumers in Jordan"

_jcm, 2025, doi:10.3390/jcm15010203_

Round 1

Reviewer 1 Report

Comments and Suggestions for Authors

This study explores the association between energy drink consumption and self-reported bruxism behaviors and temporomandibular joint pain among young adults in Jordan through a cross-sectional survey. The research question holds public health significance, the study design is generally sound, the statistical analysis is appropriate, and the conclusions are reasonably supported by the data. However, there is room for improvement in the description of methods, presentation of results, and depth of discussion. 

  1. Excluding participants with high stress to control for confounding is reasonable, but the "Discussion" section should delve deeper into how this limits the generalizability of the findings. For example, could this underestimate the moderating or mediating role of stress in the energy drink-bruxism relationship?
  2. While Table 1 provides the stress distribution, it is recommended to include a more comprehensive table of sample demographic characteristics (e.g., education level, occupation) in the main text or supplementary materials to enhance sample transparency.
  3. Excluding individuals who consume >2 cups of coffee per day is a good method to control for caffeine intake. However, consider specifying the definition of “2 cups” (e.g., volume in ml or caffeine content in mg) to enhance reproducibility.
  4. The report mentions that 26.9% of consumers “suspected” they had bruxism. It is suggested to briefly explain in the analysis or discussion how this group differs from the "confirmed" group, or to treat it as part of a sensitivity analysis.
  5. The discussion could be more specific in linking the physiological pathways through which energy drink components (e.g., the synergistic effects of taurine and caffeine) might influence bruxism, rather than relying on general statements.
  6. The study sample primarily consists of young adults aged 18-22 (88%) and excludes individuals with various health conditions. The limitations section should more explicitly state that the results may not be generalizable to older populations, those with comorbidities, or individuals from different cultural backgrounds.
  7. Some references are dated (e.g., 1992). It is recommended to supplement, where appropriate, with key studies from the last 5 years in this field, particularly regarding energy drinks and neuromuscular function.

Author Response

1. Summary

Thank you very much for taking the time to review this manuscript. Please find the detailed responses below and the corresponding revisions/corrections highlighted in red in the resubmitted files.

2. Questions for General Evaluation

Reviewer’s Evaluation

Response and Revisions

Does the introduction provide sufficient background and include all relevant references?

Yes

Are all the cited references relevant to the research?

Yes

Is the research design appropriate?

Yes

Are the methods adequately described?

Yes

Are the results clearly presented?

Can be improved

Are the conclusions supported by the results?

Can be improved

3. Point-by-point response to Comments and Suggestions for Authors

This study explores the association between energy drink consumption and self-reported bruxism behaviors and temporomandibular joint pain among young adults in Jordan through a cross-sectional survey. The research question holds public health significance, the study design is generally sound, the statistical analysis is appropriate, and the conclusions are reasonably supported by the data. However, there is room for improvement in the description of methods, presentation of results, and depth of discussion.

Comments 1: Excluding participants with high stress to control for confounding is reasonable, but the "Discussion" section should delve deeper into how this limits the generalizability of the findings. For example, could this underestimate the moderating or mediating role of stress in the energy drink-bruxism relationship?

Response 1: We thank the reviewer for this helpful suggestion. We have expanded the Discussion (Lines 253–260) to more thoroughly address how excluding participants with high stress affects the generalizability of our findings. Specifically, we discuss that the observed associations may represent conservative estimates, and we highlight the potential moderating or mediating role of stress in the relationship between energy drink consumption and bruxism behaviors.

Comments 2: While Table 1 provides the stress distribution, it is recommended to include a more comprehensive table of sample demographic characteristics (e.g., education level, occupation) in the main text or supplementary materials to enhance sample transparency.

Response 2: We thank the reviewer for this helpful suggestion. A descriptive paragraph summarizing the study sample has been added, and a new table (Table 1) has been added as well, presenting participants’ age, gender, and occupation, has been integrated into the main text to enhance transparency (Lines 167–180).

Comments 3: Excluding individuals who consume >2 cups of coffee per day is a good method to control for caffeine intake. However, consider specifying the definition of “2 cups” (e.g., volume in ml or caffeine content in mg) to enhance reproducibility.

Response 3: We thank the reviewer for this suggestion. We have added a description in the Methods section specifying that participants reported coffee consumption in terms of cups per day. Although cup volume and caffeine content were not directly measured, previous research in Jordan indicates that a typical cup of brewed coffee contains approximately 113–247 mg of caffeine depending on type and preparation method (Hammad et al., 2015). This provides context for the cutoff of >2 cups per day used to control for additional caffeine intake (Lines 142–146)

Comments 4: The report mentions that 26.9% of consumers “suspected” they had bruxism. It is suggested to briefly explain in the analysis or discussion how this group differs from the "confirmed" group, or to treat it as part of a sensitivity analysis.

Response 4: We thank the reviewer for this helpful comment. In this study, bruxism was assessed as a self-reported behavioural outcome rather than a clinically confirmed diagnosis. Participants reporting “suspected” bruxism were included alongside those reporting bruxism behaviours, as both reflect perceived parafunctional activity commonly captured in population-based research. We have clarified this point in the Discussion to emphasize that the observed associations pertain to overall self-reported bruxism behaviours and are not dependent on diagnostic certainty (Discussion, Lines 247-252).

Comments 5: The discussion could be more specific in linking the physiological pathways through which energy drink components (e.g., the synergistic effects of taurine and caffeine) might influence bruxism, rather than relying on general statements.

Response 5: We thank the reviewer for this valuable comment. In response, we have expanded the Discussion to provide a more specific description of the potential physiological pathways linking energy drink components to bruxism (Lines 272–282). The revised text details taurine’s role in modulating intracellular calcium handling and inhibitory neurotransmitter systems and explains how taurine may act synergistically with caffeine to influence neuromuscular excitability and jaw-muscle activity, providing a mechanistic rationale for the observed associations.

Comments 6: The study sample primarily consists of young adults aged 18-22 (88%) and excludes individuals with various health conditions. The limitations section should more explicitly state that the results may not be generalizable to older populations, those with comorbidities, or individuals from different cultural backgrounds.

Response 6: We thank the reviewer for this comment. The Limitations section has been revised to explicitly state that the predominance of young adults aged 18–22 years and the exclusion of individuals with medical conditions may limit the generalizability of the findings to older populations, individuals with comorbidities, or different cultural contexts (Lines 293–297).

Comments 7: Some references are dated (e.g., 1992). It is recommended to supplement, where appropriate, with key studies from the last 5 years in this field, particularly regarding energy drinks and neuromuscular function.

Response 7: We thank the reviewer for this helpful suggestion. In response, we have updated the reference list by adding eight recent studies published within the last five years, particularly focusing on energy drink, caffeine and taurine effects on neuromuscular activity and physiological effects. These additions strengthen the contemporary relevance of the discussion and mechanistic interpretation of our findings

We sincerely thank the reviewer for their thoughtful and constructive feedback, which has substantially strengthened the quality and clarity of the manuscript. We hope that the revisions adequately address their comments.

Reviewer 2 Report

Comments and Suggestions for Authors

Dear authors I read your article ‘’Prevalence of Bruxism Among Young Adult Energy Drink Consumers in Jordan”and below is my evaluation:

The topic is relevant in the current context-increasing of consumtion for enetrgy drink among young adults. Objectives are clear and well defined.

To encreasy the validity of results, you should add real time evaluation for bruxism (portable EMG devise). Because of self reported measurments you have reporting errors.

Results show a significant correlation between the frequency of energy drink consumtion and self reported bruxism behaviours.

Statistical analisis should be completed with longitudinal evaluation.Please make separation between awake bruxism and sleep bruxism and make statistical analisis for this types. Add data related with coffe consumtion and energy drink because it is possible to have cases with bouth drinks.

For discusion you should clearlly separate the published data from your own clinical experience.

Author Response

1. Summary

Thank you very much for taking the time to review this manuscript. Please find the detailed responses below and the corresponding revisions/corrections highlighted in red in the re-submitted files.

2. Questions for General Evaluation

Reviewer’s Evaluation

Response and Revisions

Does the introduction provide sufficient background and include all relevant references?

Yes

Are all the cited references relevant to the research?

Yes

Is the research design appropriate?

Can be improved

Are the methods adequately described?

Yes

Are the results clearly presented?

Yes

Are the conclusions supported by the results?

Yes

3. Point-by-point response to Comments and Suggestions for Authors

Dear authors I read your article ‘’Prevalence of Bruxism Among Young Adult Energy Drink Consumers in Jordan”and below is my evaluation:

The topic is relevant in the current context-increasing of consumption for energy drink among young adults. Objectives are clear and well defined.

Comments 1: To increase the validity of results, you should add real time evaluation for bruxism (portable EMG devise). Because of self reported measurements you have reporting errors.

Response 1: We thank the reviewer for this important observation. We acknowledge that objective real-time assessments such as ambulatory EMG or polysomnography (PSG) represent the gold standard for evaluating bruxism and would further strengthen internal validity. However, due to the large, population-based nature of the present study, bruxism behaviors were assessed using self-reported measures, which are widely employed in epidemiological research and appropriate for large-scale studies where instrumental assessment is not feasible.

The Limitations section has been revised to explicitly acknowledge the potential for reporting bias related to self-reported bruxism and to emphasize that future studies should incorporate objective monitoring tools, such as portable EMG devices, to validate and extend these findings (Lines 298–301).

Comments 2: Results show a significant correlation between the frequency of energy drink consumption and self reported bruxism behaviours. Statistical analysis should be completed with longitudinal evaluation. Please make separation between awake bruxism and sleep bruxism and make statistical analysis for this types. Add data related with coffee consumption and energy drink because it is possible to have cases with both drinks.

Response 2: We thank the reviewer for these suggestions. We acknowledge that the cross-sectional design limits causal inference, and longitudinal analyses are not possible within the current dataset. Similarly, the survey assessed overall self-reported bruxism behaviors without separate classification into awake and sleep bruxism; future studies using objective or detailed real-time monitoring could address these distinctions.

Regarding coffee intake, participants consuming more than two cups per day were excluded to control for high baseline caffeine intake. While it is possible that participants consuming ≤2 cups of coffee per day also consumed energy drinks, the study was designed to focus on energy drink consumption as the primary exposure, with coffee intake controlled rather than analyzed as a parallel variable and coffee intake within this range was not analysed as a separate covariate. A clarifying sentence has been added to the Methods (Lines 147–149) to reflect this.

Comments 3: For discussion you should clearly separate the published data from your own clinical experience.

Response 3: We thank the reviewer for this suggestion. A clarifying sentence has been added to the Discussion (Lines 239–240) to explicitly state that interpretations are based on the study findings and previously published evidence, and that personal clinical experience has not been included.

We sincerely appreciate the reviewer’s thoughtful feedback. Although the study design and scope limited additional analyses, we have clarified methods, highlighted limitations, and contextualized the findings. We hope these revisions make the manuscript clearer and more transparent.